# Predictive coding feedback results in perceived illusory contours in a recurrent neural network

**Zhaoyang Pang**
CerCo CNRS, UMR 5549 &
Université de Toulouse

**Bhavin Choksi**
CerCo CNRS, UMR 5549 &
Université de Toulouse

**Callum Biggs O'May**
CerCo CNRS, UMR 5549

**Rufin VanRullen**[*]
CerCo CNRS, UMR 5549 &
ANITI, Université de Toulouse

## Abstract

Feedforward Convolutional Neural Networks (CNNs) have made great strides in solving computer vision tasks. However, these networks only roughly mimic human visual perception [1, 2]. For example, a recent report [3] shows that such neural networks do not perceive illusory contours (e.g., Kanizsa squares [4]) in the way that humans do. Physiological evidence suggests that the perception of illusory contours could involve feedback connections in the visual cortex [5, 6], which are lacking in feedforward networks [7, 8]. Would recurrent feedback neural networks perceive illusory contours like humans? In this work we address this issue by equipping a deep feedforward convolutional network with brain-inspired recurrent dynamics implementing a "predictive coding" strategy: at each layer of the hierarchical model, generative feedback "predicts" (or reconstructs) the pattern of activity in the previous layer, and the reconstruction errors are used to iteratively update the network's representations across timesteps. The neural network was first pretrained on the CIFAR100 natural image dataset in an unsupervised way with a reconstruction objective. Then, a classification decision layer was added and the model was finetuned on a form discrimination task: squares vs. randomly oriented inducer shapes (no illusory contour). Finally, the model was tested with the unfamiliar "illusory contour" configuration: inducer shapes oriented to form an illusory square. Compared with the feedforward baseline, the iterative "predictive coding" feedback resulted in more illusory contours being classified as physical squares across timesteps. The illusory contour was measurable in the luminance profile of the image reconstructions produced by the model. In other words, the model was not merely confused by the novel configuration, but behaved as if a shape had truly been presented, similar to the human version of the illusion.

## 1 Architecture

We construct a 3-layered hierarchical stacked autoencoder with 3 feedforward encoding layers $e_n$ ($n \in 1, 2, 3$) and 3 corresponding feedback decoding layers $d_{n-1}$ (see Figure 1). When considering only the encoding layers, the network can be viewed as a standard feedforward convolutional neural network. To guide the implementation of the feedback connections, we follow the principles of "predictive coding" as introduced by Rao and Ballard [9], a popular framework in neuroscience for characterizing cortical function: in the hierarchical network, the higher layers try to predict the

---

[*]correspondence: rufin.vanrullen@cnrs.fr

2nd Workshop on Shared Visual Representations in Human and Machine Intelligence (SVRHM), NeurIPS 2020.

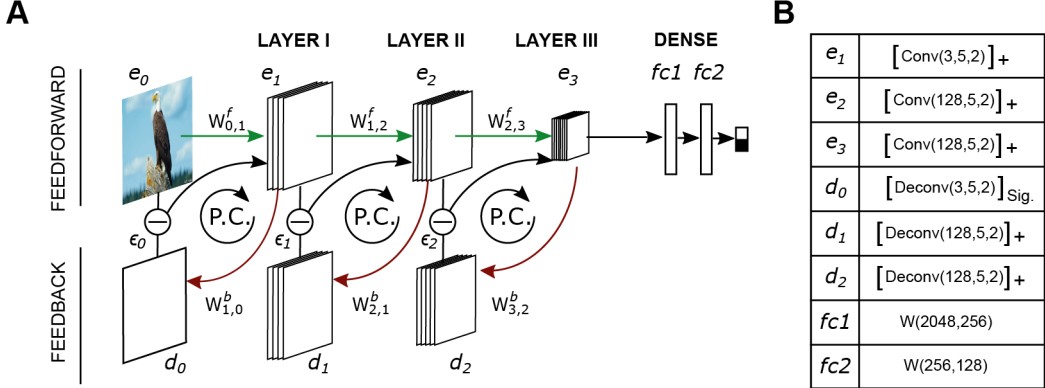

**A**

FEEDFORWARD

LAYER I    LAYER II    LAYER III    DENSE
                                     fc1  fc2

$e_0$    $e_1$    $e_2$    $e_3$

$W^f_{0,1}$    $W^f_{1,2}$    $W^f_{2,3}$

$\epsilon_0$  P.C.  $\epsilon_1$  P.C.  $\epsilon_2$  P.C.

FEEDBACK

$W^b_{1,0}$    $W^b_{2,1}$    $W^b_{3,2}$

$d_0$    $d_1$    $d_2$

**B**

| | |
|---|---|
| $e_1$ | $\big[\text{Conv}(3,5,2)\big]_+$ |
| $e_2$ | $\big[\text{Conv}(128,5,2)\big]_+$ |
| $e_3$ | $\big[\text{Conv}(128,5,2)\big]_+$ |
| $d_0$ | $\big[\text{Deconv}(3,5,2)\big]_{\text{Sig.}}$ |
| $d_1$ | $\big[\text{Deconv}(128,5,2)\big]_+$ |
| $d_2$ | $\big[\text{Deconv}(128,5,2)\big]_+$ |
| $fc1$ | $W(2048,256)$ |
| $fc2$ | $W(256,128)$ |

Figure 1: **Three-layered auto-encoder equipped with a predictive coding strategy.** A. Network architecture. The architecture consists of a main body and a classification head (or dense layers). For the main body, the predictive coding strategy is implemented in stacked auto-encoders with three feedforward encoding layers ($e_n$) and three generative feedback decoding layers ($d_n$). Reconstruction errors ($\epsilon_n$) are computed and used by the proposed predictive coding updates which are denoted by 'P.C.' loops. Dense layers are added on top of the structure to implement a binary classification task. B. Table of parameters. Each encoding layer is a combination of a convolution layer and a ReLU nonlinearity with parameters Conv(channels, kernel size, stride). Decoding layers consist of a deconvolutional layer with ReLU non-linearity, except for layer $d_0$, which uses a Sigmoid activation function, in order to compare with the input picture with pixel values ranging from 0 to 1. After flattening the output of the last convolutional layer, and going through a batch normalization function, two dense layers with a structure of Weight(in features, out features) project to the binary decision layer.

activity of the lower layers and the errors made in this prediction are then utilized to update their activity. For a given input image, we initiate the activations of all encoding layers with a feedforward pass—that is, activity at time zero can be viewed as a feedforward baseline for the model. Then over successive recurrent iterations (referred to as timesteps $t$) we update both the decoding and encoding layer representations using the following equations:

$$d_n(t+1) = \big[W^b_{n+1,n}e_{n+1}(t)\big]_+$$

$$e_n(t+1) = \beta_n\big[W^f_{n-1,n}e_{n-1}(t+1)\big]_+ + \lambda_n d_n(t+1) + (1 - \beta_n - \lambda_n)e_n(t) - \alpha_n\frac{\partial\epsilon_{n-1}(t)}{\partial e_n(t)},$$

$$(1)$$

where $W^f_{n-1,n}$ denotes the feedforward weights connecting layer $n-1$ to layer $n$, and $W^b_{n+1,n}$ denotes the feedback weights from layer $n+1$ to $n$. The parameters $\beta_n$, $\lambda_n$ and $\alpha_n$ act as balancing coefficients for the feedforward, feedback and error correction terms respectively, and they are treated as hyperparameters of the network (for the present experiments these hyperparameter values were fixed to $\beta = 0.2$, $\lambda = 0.1$ and $\alpha = 100$).

All the weights $W$ are fixed during the iterations defined by Eq 1. They are optimized over successive batches of natural images (see Methods) to minimize the total reconstruction error $L$ (Eq 2)–an unsupervised objective in accordance with the principles of the predictive coding theory. In Eq 2, $N$ is the number of layers (here, $N = 3$) and $\epsilon_n$ denotes the reconstruction error at layer $n$, defined as the mean squared error between the representation $e_n$ and the corresponding prediction $d_n$.

$$L = \sum_t\sum_{n=0}^{N-1}\epsilon_n(t) = \sum_t\sum_{n=0}^{N-1}\|e_n(t) - d_n(t)\|_2 \qquad (2)$$

Intuitively, each of the four terms in Eq 1 contributes different signals to a layer: (i) the feedforward term (controlled by parameter $\beta$) provides information about the (constant) input and changing representations in the lower layers, (ii) the feedback term (parameter $\lambda$) guides activations towards their representations from the higher levels, (iii) the memory term helps to retain the current representation

over successive timesteps, and (iv) the error correction term (controlled by parameter $\alpha$) corrects representations in each layer such that their predictions can better match the preceding layer. Together, the feedback and error correction terms fulfill the objective of predictive coding as laid out by Rao and Ballard [9].

We hypothesize that this architecture, due to its training on natural images containing numerous object contours, and its generative properties that iteratively modify lower-level representations to match higher-level descriptions, could be sensitive to "illusory contours", much like human vision. A similar architecture and its application to robust image recognition on the large-scale ImageNet dataset are described in a companion paper [10].

## 2   Prior work

Several studies have investigated the perception of illusory contours in deep artificial neural networks. Baker et al. [3] tested whether a feedforward deep convolutional network -AlexNet [11]- could interpolate illusory contours between inducing elements in the same way as humans. They employed a so-called classification image technique [12] to give insight into which regions in a given image are important to an observers' perceptual decisions. Applying this technique to human subjects found that the region between inducers is influential in humans' perceptual decisions [12]. However, applying the same technique to a feedforward network showed that it did not appear to represent illusory contours – instead, it relied only on the orientation of inducing elements to make its decisions [3], illustrating a fundamental difference between the visual representations formed by deep networks and humans.

The above-described classification image technique used an external noise probe to infer the network's representation for illusory contours. Kim et al. [13] on the other hand directly examined the activations of intermediate layers in feedforward neural networks. They computed the cosine similarity between layers' activations for illusory contours and for physical shapes. Their results suggest illusory contours are closer or more similar to physical contours in the network's representational space than control non-illusory shapes. However, this effect was more pronounced in the network's higher layers, which may be inconsistent with physiological evidence suggesting that the perception of illusory contours originates in early visual cortex [5]. Moreover, the similarity in the layers' activations only indirectly suggests that the network is truly *perceiving* illusory contours.

In addition to feedforward neural networks, Lotter et al. [14] investigated a type of recurrent neural network, PredNet [15], which is similar but not identical to the current network. They compared the response properties of single units in PredNet to neuronal recordings in the primate visual cortex and showed similar dynamic responses in the presence of illusory contours. Nevertheless, their research only focused on the neuronal/layer level, and could not directly conclude that the model 'perceived' illusory shapes. Also, it should be noted that Lotter et al. trained their network on dynamic stimuli to predict future from past video frames, and consequently tested illusory contours in a dynamic situation—different from the human version of the illusion which can be experienced on static images.

## 3   Methods and Results

### 3.1   Training procedure

The network's training included two stages (both with 10 timesteps): pretraining and finetuning. The pretraining was conducted in an unsurpervised way with a reconstruction objective, wherein both feedforward and feedback weights were optimized over the CIFAR100 natural images dataset [16]. This was done to learn a hierarchy of relevant features to describe each natural image, as well as the corresponding generative pathway to reconstruct images from their features. The finetuning stage was done in a supervised way, with a classification decision layer added to the network to perform a binary form-discrimination task. Specifically, we generated 4 classes of stimuli: Square (physical squares), Random (randomly oriented inducers), All-in (illusory contours, in which the inducers faced each other), and All-out (a control condition, in which all inducers were facing away from each other). These stimuli varied in luminance (range [0 - 1], for both background and inducer luminance), size, and position (Figure 2A). For each image, Gaussian noise was added (with variance randomly sampled from 7 noise levels) to render the task non-trivial and to avoid overfitting. Notably, during finetuning, the network was only presented with the Square class and the Random class, and trained

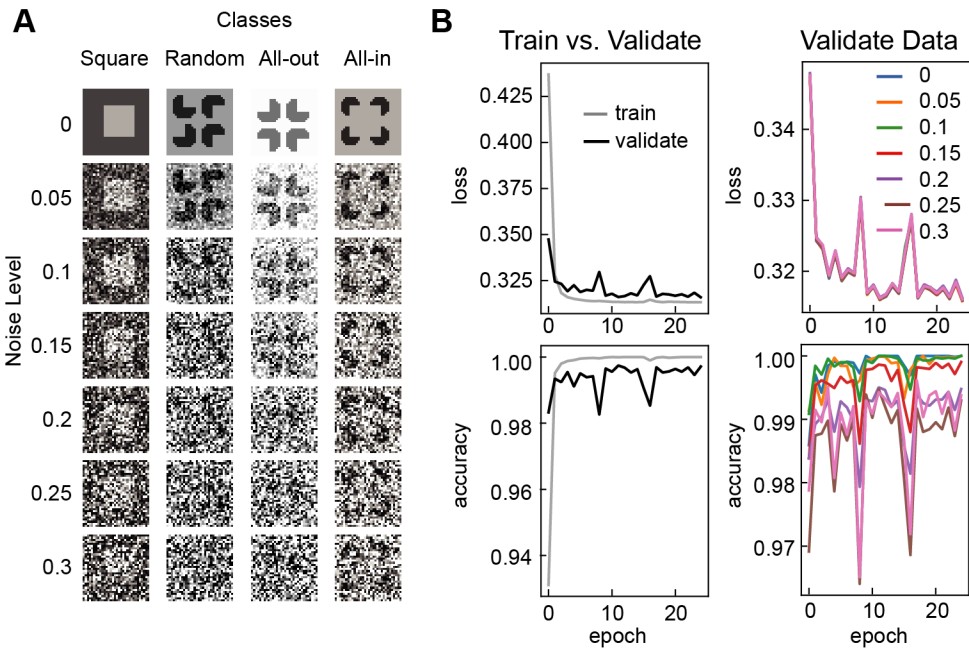

Figure 2: **Finetuning dataset and results.** A. Sample training and testing images for 7 different levels of Gaussian noise: Square, Random, All-out, and All-in. All stimuli varied in luminance (both background luminance and inducer luminance), size (square sidelength or distance between inducers), and position. B. Training results. Left: Average training loss and accuracy results over all noise levels. Right: Same, split by noise level.

to discriminate between them. In other words, no illusory contour (All-in) or control (All-out) images were presented in this phase. The weights of the whole network were finetuned for 25 epochs, after which it was tested with a distinct validation set. We performed three pretrainings (each with different randomly initialized weights), and then each of these pretrainings was used to fine-tune 3 networks. The test results reported are averaged over the resulting 9 networks. Figure 2B shows the training results for an example network, for which the accuracy on the validation set was around 0.997, with higher values for lower Gaussian noise level.

### 3.2 Illusory contour perception

After pre-training on natural images and fine-tuning on simple shapes, the network could discriminate between physical squares and randomly oriented inducers in any configuration (except the two critical configurations, All-in and All-out, which had not been seen during training). During testing, all four classes were presented to the network, with All-out (control class) and All-in (illusory contours) as novel stimulus configurations formed by familiar inducers. Our hypothesis was that the All-in configuration would be classified as a square in a number of trials (illusory contour perception), whereas the All-out configuration would be more likely to be classified as random inducers.

For each class, we thus measured the probability of the image being classified as a square (Figure 3, results averaged over 9 networks). Results at timestep 0 correspond to the output of a standard feedforward convolutional network. The feedforward network appears to classify images based on low-level, local information, as all the pacmen-made patterns (Random, All-out, All-in) are recognized as non-square shapes. However, over timesteps, the network begins to recognize the All-in condition (the illusory contour) as a square, at a much higher rate than the All-out and Random conditions. After 50 timesteps, the likelihood of reporting a square (an "illusory contour perception") has increased by more than 50-60% for the All-in condition (depending on noise level), whereas it remains around 20-50% for the other conditions. Although it does not reach 100%, we suggest that even humans would not systematically categorize inducers as squares—in fact, despite 'seeing' the illusion, we easily recognize that there is no actual square in the image. As the noise level increases (Figure 3B), the likelihood of reporting a square for the All-in condition increases, along with the

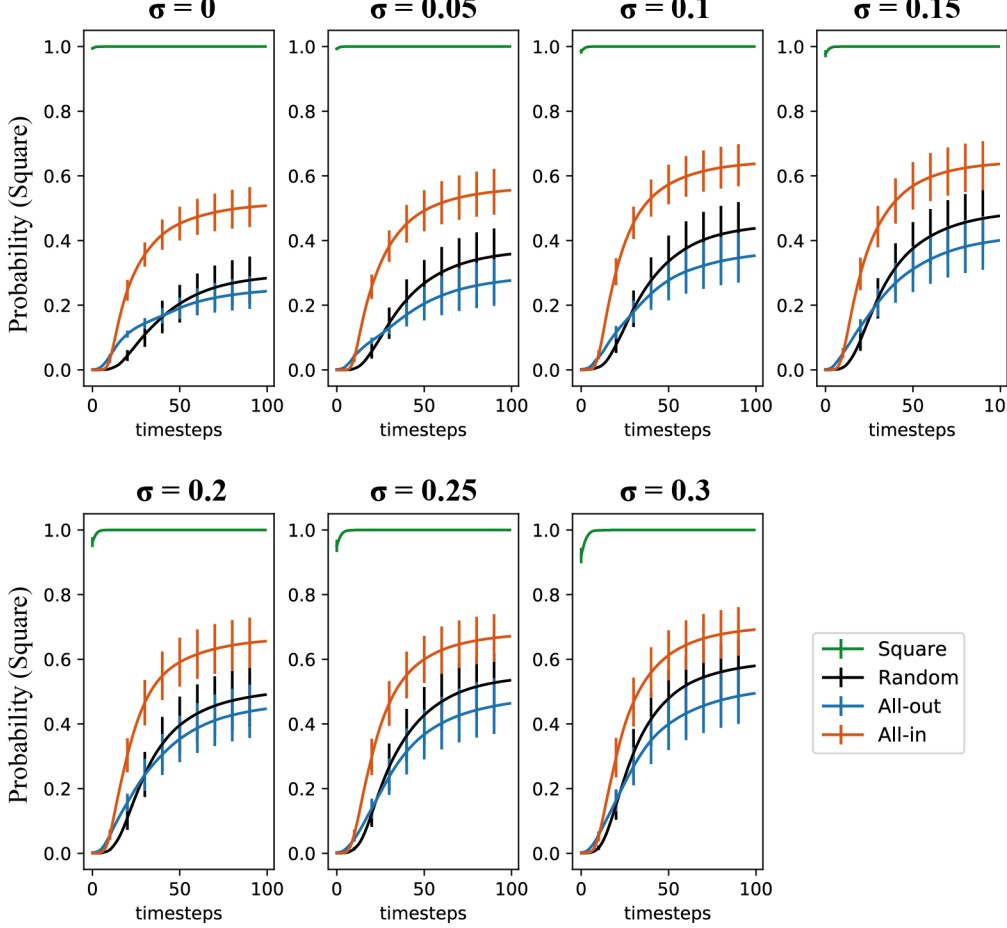

Figure 3: **Classification results.** The probability of a "square" report over recurrent predictive coding iterations (timesteps). Each panel shows a different noise level. Feedback iterations increase the likelihood of a "square" report, especially for the All-in condition. Results are averaged over 9 networks and error bars represent standard error of the mean (SEM).

likelihood for the All-out and Random conditions: it appears that the network becomes generally more 'confused'. Still, the illusory contour effect (higher 'square' classification for All-in than All-out stimuli) does not disappear, even at high levels of noise.

Although the network classified more illusory contours as squares, it remains hard to draw the conclusion that the network could really "see" illusory contours. A major advantage of the predictive coding framework is that we can inspect the reconstructions at each layer. Thus, we examined the image reconstructions produced by the model. Figure 4B displays two examples of reconstructed images at timestep 100. Compared with the original images (top row) whose noise standard deviation is 0.1, the reconstructed images are denoised by the network, and the illusory contour shapes are even clearer. To quantify the illusion "perceived" by the network, we inspected the luminance profile of the reconstructions: for each image we computed a "Figure-ground luminance difference" (FG). Given the expected position of the illusory contour (the position that the square would have occupied if it had been real instead of illusory), along each of the four cardinal axes we took two pixels inside (red in Figure 4A) and two pixels outside the square (green in Figure 4A). We computed the average difference between the pixel luminance values inside and outside. A polarity factor (-1 or 1) multiplied this measure, to take into account the different configurations: dark inducers (polarity=1) are expected to produce lighter illusory shapes, light inducers (polarity=-1) to produce darker ones. The constructed FG measure is zero in the original All-in images (since it is measured in the background between the inducers), but should be positive in the image reconstructions whenever an illusory contour is perceived. Figure 4C shows how this FG value changes over predictive coding

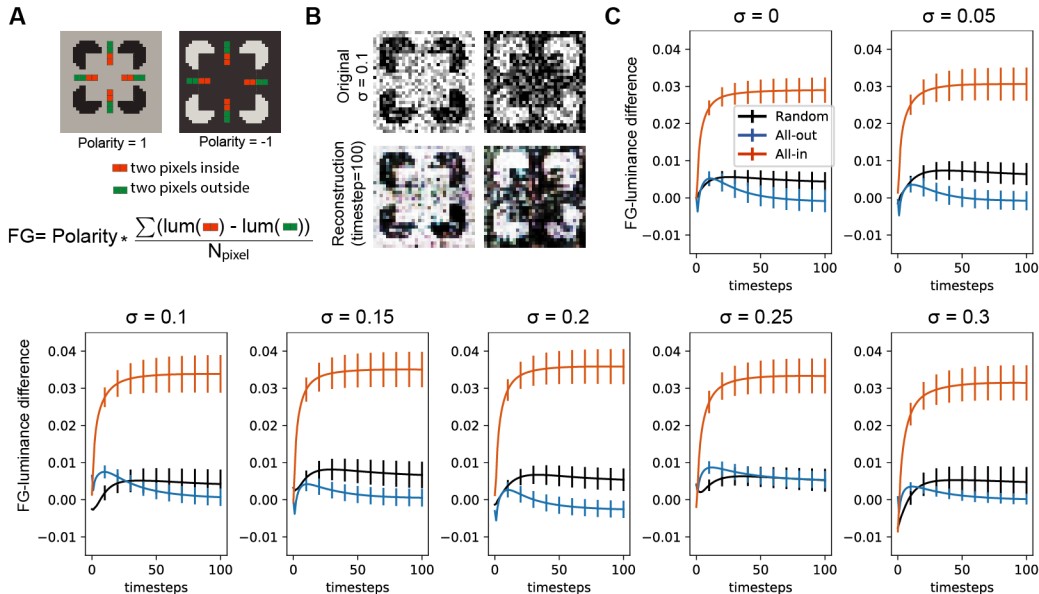

Figure 4: Quantification of illusory countour perception in the neural network. A. Computation of the "FG" value measuring the figure-ground luminance difference. B. Two examples of illusory contour (All-in) stimuli, and their corresponding reconstructions from the network at timestep 100. C. The FG-luminance difference for classes Random, All-out, and All-in over 100 timesteps (zero is absence of illusion, larger values mean more illusion is perceived). Results are averaged over 9 networks and error bars represent for SEM.

iterations, for illusory contours (All-in class) and the other two non-illusory shapes (All-out and Random classes). For the All-in class, after 10 time steps the value is consistently higher than zero, and 5 to 10 times higher than the FG value measured for the control (All-out) or the random inducer classes. For comparison, the FG value for the physical square would range from 0.3 to 0.6 (not shown here). Remarkably, these results are consistent with physiological evidence showing that neural responses for illusory contours in monkey V1 are higher than similar non-illusory images, but lower than physical squares [5].

## 4 Discussion

The purpose of this study was to test whether a recurrent feedback neural network could perceive illusory contours (Kanizsa squares) in a similar manner to humans. We augmented a feedforward CNN with predictive coding recurrent dynamics so that, unlike other related work [14, 3, 13], we were able to (i) analyse explicit classification decisions (square vs. inducers) and (ii) visualize reconstructed inputs from the model's viewpoint. We found that, compared to a feedforward baseline, the recurrent dynamics led the network to perceive more and more illusory contours. Further, by inspecting the network's reconstructions, we found evidence of modulations of the perceived luminance profiles in the expected direction for illusory shapes, suggesting that the network was truly "perceiving" the contours. In summary, we provide evidence that brain-inspired recurrent dynamics can lead networks to perceive illusory contours like humans.

## Broader Impact

This work is at the intersection of machine learning and computational neuroscience, and we leverage each field to better understand the other. Fostering further integration and collaboration between the two has the potential to advance both significantly. In this work we primarily contribute to the ongoing goal of better understanding biological neural systems, which has wide-ranging potential societal benefits including potential improvements in medical care. The risk of this aspect of the work having harmful consequences seems relatively low.

We also demonstrate that taking inspiration from the brain can help develop artificial intelligence systems which share similarities with human perception, potentially contributing to the development of systems which better mirror human behavior. Given the rapidly progressing integration of machine learning systems into many everyday tools and the ongoing discussions surrounding transparency, accountability, and explainability, the possible positive outcomes include helping us better understand the tools being deployed. On the other hand, although this work is demonstrating a relatively low-level effect (the perception of visual illusion), in general this approach may carry risks that include enabling malicious groups to impersonate humans with AI.

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
