# OpenReview forum: "Predictive coding feedback results in perceived illusory contours in a recurrent neural network"
_NeurIPS.cc/2020/Workshop/SVRHM — SVRHM@NeurIPS Poster_

### Official Review · AnonReviewer2 · 2020-10-29
**An account for the Kanizsa square illusion in a predictive coding network**

**Rating:** 9
**Confidence:** 4

**Review:**

This paper reproduces the perception of illusory contours in the Kansizsa square illusion (https://en.wikipedia.org/wiki/Illusory_contours) in a predictive coding model of the visual system. The model, inspired by (Rao and Ballard 1999), is a hierarchical autoencoder network where the higher layers predict the activity of the lower layers. The paper shows that (1) a decoder trained on the output of the network confuses the Kansizsa square with a real square (consistent with the illusion in humans) (2) the reconstructed image from the last layer of the network have illusory contours (consistent with physiology).

This work is very interesting from a neuroscience standpoint, and it also paves the way to innovative ML architectures inspired by models of the visual system from traditional theoretical neuroscience. The authors say: "A similar architecture and its application to robust image recognition on the large-scale ImageNet dataset are described in a companion paper (under review).", which sounds very promising.

The paper is extremely well written and easy to follow, and the experiments and model are clearly described and rigorous.

Given these qualities and the fit with the workshop theme, I am nominating this paper for the award of the workshop.

Comment:
- In equation 1, what is \epsilon?

**Bio Award:**

Yes, paper should be nominated as I have given it a high score and it is also relevant to the award (presents a biologically-driven generative model).

---

### Official Review · AnonReviewer3 · 2020-10-29
**Nice task, encouraging results, but this is still far from a sufficient model of human illusory contour perception.**

**Rating:** 6
**Confidence:** 4

**Review:**


This work demonstrates the detection of a Kanizsa's square and its associated illusory contours in a predictive-coding like CNN trained on the detection of non-illusory squares. The modeling approach is described in more detail in another submission to the SVRHM 2020 workshop. The network is trained on CIFAR-100 (only on reconstruction, no classification cost) and is then finetuned on the task of discriminating between images with randomly oriented inducers ("pacmans") and images with a (non-illusory) square. Both conditions had varying shape sizes, locations, and polarities, as well as added noise. When tested with aligned pacmans (`All-in'), the recurrent predictive coding processing induced a slight bias (a choice probability of 0.05-0.1) towards detecting the square class. This effect appeared with a considerably smaller magnitude for images with randomly-oriented or outward-facing pacmans, and becomes more evident for images with greater noise. In addition, the reconstructed pixels showed a (weak) brightness difference across the edges of the illusory square. This effect is consistent with a representation of an illusory contour.

Overall, I think that this is an encouraging finding. It provides another piece of evidence in favor of predictive coding top-down connections as a necessary component in deep neural network-based models of vision.

My main concern with this study is that the observed model behavior (i.e., classifications, Figure 2) is not really consistent with human behavior. First, the behavioral bias towards detecting a square in the "all-in" condition is **tiny** compared to human behavior (i.e., humans always see Kanizsa's squares as squares). Secondly, the network is more inclined to see squares in the high-noise conditions, whereas for humans, the illusory square seems to be more evident in the no-noise or low-noise conditions (just judging from my own perception of the rightmost column of Figure 1A).

 I think that both discrepancies might be related to the way the network was trained (discriminating between random pacmans and squares). An alternative, potentially better training approach would be to finetune the network to detect squares, with random pacmans as *non-informative* distractors which may appear beside or on top of the squares. A larger input image (e.g., 224 x 224) would make this more practical. Trained on this task, the network will not learn that the detection of pacmans entails that there's no square. Right now, it seems that as the pacmans become visible, the network is driven to respond 'no square', leading to human-inconsistent behavior.

Once qualitatively human-like behavior is achieved by improving the model, the authors can strengthen their evidence by quantifying the model-human consistency. This can be done by presenting human subjects with the test stimuli, asking them to perform a simple `square/non-square' classification. If the model captures the human perceptual mechanism applied in this task, its classification probabilities should predict human classification probabilities across stimulus conditions.

Minor comments:
* Figure 2A inadvertently and incorrectly suggests that the "All-in" and the "All-out" conditions are confounded by polarity.
* The details about the network architecture and the training procedure are insufficient.

---

### Official Review · AnonReviewer1 · 2020-10-30
**Missing crucial comparisons**

**Rating:** 3
**Confidence:** 4

**Review:**

The paper 'Predictive coding results in perceived illusory contours in a recurrent neural network' tests the idea that adding feedback connections between layers of a deep feedforward convolutional neural network that predicts activations in the previous layer, would lead to learned representations suitable for recognizing illusory shapes in images.

The idea of adding recurrent feedback processing to a model to recognize illusory shapes is well motivated by neurophysiological evidence. However, I have a few concerns in accepting the results of this paper --

Major concerns:
1. To really claim that the addition of feedback connections has helped recognize illusory shapes, one needs to compare it against a purely feedforward model. It has been observed before that a network with feedback/recurrent connections might not do much better than a deeper purely feedforward network on a visual task that is believed to involve recurrent processing in the brain (see supplementary figure 4 in Tang, H., Schrimpf, M., Lotter, W., Moerman, C., Paredes, A., Caro, J. O., ... & Kreiman, G. (2018). Recurrent computations for visual pattern completion. Proceedings of the National Academy of Sciences, 115(35), 8835-8840.)
2. It would be great to see what the classification accuracy is for all the four conditions in the test set. It looks like the 'all-in' condition is being classified as 'non-square' mostly, which is not encouraging. If the 'all-in' condition is not being classified as a square at all, I see little incentive to look at further analyses.
3. Although I liked the figure-ground luminance analysis, I thought the results lacked a crucial comparison. How does this measure look for original images? If I have understood it correctly, the original 'all-in' images have added noise and hence can't have 'zero' value for this measure as the authors claim.

---

### Public Comment · ~Zhaoyang_Pang1 · 2020-12-04
**General response to reviewers**

We are thankful to all the reviewers for taking the time and care to provide such useful suggestions and comments. Some of these suggestions will require further exploration in future studies, due to time limitations. Many of the comments, however, have been addressed to improve the current revised paper. We have made the following changes:

1. To emphasize that the feedback in the connected network plays a key role in the predictive coding strategy, we added the word “feedback” in the title.
2. For the Architecture section, a schematic picture (Figure 1) of the network was added to aid understanding. In the caption text for Figure 1, we described how different layers are combined to construct the architecture, as well as their corresponding parameters. We also added equation 2 to explain how we compute the reconstruction error ε.
3. We modified our algorithm for generating the dataset used during the fine-tuning stage, since some factors like the luminance values for background and foreground  figures had not been chosen in an evenly distributed range. That was likely the reason why our network had an initial bias towards the “All-out” Control class. In the revised paper, by using properly matched luminance statistics for all classes in the dataset, the bias was eliminated. As a result, the three pacman-made classes (“Random”, “All-out”, “All-in”) now show nearly the same classification accuracy at the very early timesteps (See Figure 3).
4. In the previous version, our results were obtained based on only one network (weights were randomly initialized). To reinforce confidence in our results, we now perform repeated experiments with different random initializations (9 networks) and average our results accordingly. The corrected Figure 3 shows the results for probability of square classification (averaged over nine networks). The new results solve the first question raised by Reviewer 2. That is,  the network gets more “confused” with higher noise level, which may be more consistent with human perception.

---

### Decision · Program_Chairs · 2020-11-02

Accept (Poster)